# Composite and Surface Functionalization of Ultrafine-Grained Ti23Zr25Nb Alloy for Medical Applications

**DOI:** 10.3390/ma13225252

**Published:** 2020-11-20

**Authors:** Mateusz Marczewski, Mieczysława U. Jurczyk, Kamil Kowalski, Andrzej Miklaszewski, Przemysław K. Wirstlein, Mieczysław Jurczyk

**Affiliations:** 1Institute of Materials Science and Engineering, Poznan University of Technology, Jana Pawła II 24, 61-138 Poznan, Poland; kamil.kowalski@put.poznan.pl (K.K.); andrzej.miklaszewski@put.poznan.pl (A.M.); mieczyslaw.jurczyk@put.poznan.pl (M.J.); 2Division of Mother’s and Child’s Health, Poznan University of Medical Sciences, Polna 33, 60-535 Poznan, Poland; mjur@poczta.onet.pl; 3Department of Gynaecology and Obstetrics, Division of Reproduction, Poznan University of Medical Sciences, Polna 33, 60-535 Poznan, Poland; abys@wp.pl

**Keywords:** Ti alloys, 45S5 Bioglass, metal matrix composites, electrochemical deposition, cell proliferation

## Abstract

In this study, the ultrafine-grained Ti23Zr25Nb-based composites with 45S5 Bioglass and Ag, Cu, or Zn additions were produced by application of the mechanical alloying technique. Additionally, the base Ti23Zr25Nb alloy was electrochemically modified in the two stages of processing: electrochemical etching in the solution of H_3_PO_4_ and HF followed by electrochemical deposition in Ca(NO_3_)_2_, (NH_4_)_2_HPO_4_, and HCl. The in vitro cytocompatibility studies were also done with comparison to the commercially pure titanium. The established cell lines of Normal Human Osteoblasts (NHost, CC-2538) and Human Periodontal Ligament Fibroblasts (HPdLF, CC-7049) were used. The culture was conducted among the tested materials. Ultrafine-grained titanium-based composites modified with 45S5 Bioglass and Ag, Cu, or Zn metals have higher biocompatibility than the reference material in the form of a microcrystalline Ti. Proliferation activity was at a stable level with contact with studied materials. In vitro evaluation research showed that the ultrafine-grained Ti23Zr25Nb-based composites with 45S5 Bioglass and Ag, Cu, or Zn additions, with a Young modulus below 50 GPa, can be further used in the biomedical field.

## 1. Introduction

β-type titanium alloys are promising materials based on the current research studies in terms of their biological properties. Currently published results of TiNb, TiZrNb, TiNbTa, TiNbSn, TiNbHf, TiMoNb, TiMoTa, TiNbTaZr, TiNbZrAl, TiNbZrSi, and TiNbZrTaSiFe confirm their high biocompatibility and makes them interesting for further research studies and development in the biomedical field [1,2,3,4,5,6,7,8,9,10,11]. Zr, Nb, and Mo (with titanium-based alloys containing these elements) are biocompatible, and their cytotoxicity is lower than that for pure titanium [12].

The modification of titanium with calcium phosphate (CP) is a widely used method to modify the properties of titanium biomaterials. There are numerous methods of doing so and diverse research studies proving the utility of this type of modification (especially considering the enhancements of biological properties as excellent biocompatibility). CP could be used in the form of coatings. It has revealed the possibility of increasing the biological properties of the material. Various deposition methods can be applied to obtain CP layers on the titanium substrate, including plasma spraying, electrophoretic deposition, and electrochemical deposition (ED). The electrodeposition methods offer many advantages compared to other methods. During the ED process, it is possible to control the chemical composition of layers and their thickness. ED also allows covering irregular shapes and porous structures of implants. The formation of multilayer and two-layer composites of hydroxyapatites (HA) with titanium oxide leads to increased cell proliferation [13]. The nano titanium oxides as a monolayer also can form the surface with interesting properties such as high osteointegration properties [14]. Hydrothermal treatment leading to calcium titanate formation increases apatites’ formation on the titanium surface in the body fluids, enhancing its biocompatibility [15]. Recently, a high point of interest was also focused on the doping of this type of coatings. Calcium phosphates modification with silver, silicon, strontium, magnesium, zinc, cobalt, and multiwalled carbon nanotubes leads to the formation of the coating with no toxicity against fibroblast and high potential because of the improved osteointegration [16,17,18,19]. The production of titanium-based composites containing calcium phosphates also provides a faster growth of apatites on the material and high biocompatibility with excellent mechanical properties [20,21].

There is also high interest in biomaterials engineering for the use of bioglasses such as 45S5 Bioglass [22]. The bioglass can also be additionally modified with the incorporation of many different elements. The addition of niobium in 45S5 Bioglass could stimulate the regeneration of the bone tissue [23,24]. The bioglass properties could also be further improved with the addition of selenium, magnesium, zinc, and hydroxyapatite [25,26]. The addition of graphene oxides can lead to both enhancing the biocompatibility and antibacterial properties [27].

45S5 Bioglass can be used as a coating to improve titanium and titanium-based alloys further, mainly their biocompatibility [28]. 45S5 Bioglass is added as a ceramic component of metal-matrix titanium-based composites. This type of composites could be easily produced with a mechanical alloying technique leading to the further improvement of composite properties and high cytocompatibility compared to commercially available pure titanium [29]. The properties of biomedical materials could also be enhanced with a different grade of bioglass as S53P4 Bioglass [30]. This type of modification was proved to be efficient in terms of β-type titanium alloys. Both hydroxyapatite and 45S5 Bioglass have proved to enhance many different alloys’ biocompatibility as Mg-based alloys [31,32]. The formation of calcium hydroxide on the titanium surface is also beneficial because of their antibacterial and osseointegration properties [33,34]. The same increase in the biocompatibility could have also been achieved with the production of other calcium-decorated coatings [35].

One of the most popular methods of modifying β-type titanium to increase their osteointegration properties is foams’ formation. This method was successfully used in the alloy containing niobium and zirconium using different foaming agents [36,37,38]. The foam production could also be applied in the titanium-based composites such as TiNb-HA composites [39]. It is also possible to produce porous coatings to increase bone tissue ingrowth into the modified implant [40].

The synthesis of bulk ultrafine-grained Ti-based materials is possible using a mechanical alloying approach allowing the refinement of powders to even the nanometer scale with its further consolidation [29,31,32,41,42]. The synthesis of ultrafine-grained β-type titanium alloys might enhance the properties of materials used in medicine. One of these alloys is Ti-Zr-Nb alloys with Nb and Zr used as the β-stabilizers [41,42]. These elements’ biocompatibility is higher than that of commercially used aluminum and vanadium in the commercially available Ti6Al4V alloys [12,43]. The production of alloys with a single-phase Ti(β) structure is possible with the alloys as Ti30Zr17Nb, Ti23Zr25Nb, Ti30Zr26Nb, Ti22Zr34Nb, and Ti30Zr34Nb even after consolidation in the temperature as low as 600 °C [42].

Ultrafine-grained Ti-based materials, due to exceptional physicochemical, mechanical, and biological properties, are considered as the next-generation biomaterials [44,45]. The main research task is to adjust the microstructure and chemical composition, leading to increased cell proliferation among the osseointegration of potentially produced implants. Proliferation can be improved by the chemical composition of materials and its surface properties with roughness included. However, the roughness of the medical product can lead to undesirable pathogens colonization [46]. That leads to other goals in enhancing the material as the limitation of these types of phenomena. As one possible way of doing so, silver particles were already studied in these applications [47]. The proposed production method with mechanical alloying led to the formation of the β-type titanium alloys with Nb and Zr with the mean average grain size of 1.2 to 1.7 µm, which proved it to be the ultrafine-grained material [42].

The structure and corrosion resistance as well as the wettability properties of surface-modified material were also examined in this work. The mechanical properties of Ti23Zr25Nb-based materials were summarized with commercially available titanium (rod Grade 2), arc-melted microcrystalline Ti18Zr24Nb, and Ti23Zr25Nb-based foam used as the reference samples to present and analyze their relations. Biocomposites formation was considered a much more effective way of Young modulus limitation than the foam formation. The significant reduction in elastic modulus is a way of reducing the risk of the stress shielding effect. Moreover, increased osseointegration is possible because of the porosities in these materials [48]. The influence of chemical composition on the biological properties was investigated. Additionally, the possibilities of electrochemical surface modification for a further improvement of osteointegration and the cytocompatibility properties of ultrafine-grained Ti23Zr25Nb alloy were introduced, and the biological properties of the so modified materials were revealed. The synthesized β alloy, Ti23Zr25Nb, has interesting properties for biomedical application.

In vitro, a cytocompatibility test was done to assess the toxicity and the interaction process with the proliferation activity in all the tested materials, including those functionalized with surface modification and composite formation. The established cell line of Normal Human Osteoblasts (NHost) and Human Periodontal Ligament Fibroblasts (HPdLF) was cultured among all tested materials.

## 2. Materials and Methods

The present work covers the research results for the surface and composite functionalized Ti23Zr25Nb (at.%) alloys.

In this study, synthesized materials are denoted as follows:
A0annealed titanium rod (Grade 2)B0Ti23Zr25Nb—cold pressed and sintered at 800 °C for 30 minB1*Ti23Zr25Nb—electrochemically etchedB1Ti23Zr25Nb—electrochemically deposited calcium and phosphorus-riched coatingB2Ti23Zr25Nb—9 wt.% 45S5 BioglassB3Ti23Zr25Nb—9 wt.% 45S5 BG–1 wt.% AgB4Ti23Zr25Nb—9 wt.% 45S5 BG–1 wt.% CuB5Ti23Zr25Nb—9 wt.% 45S5 BG–1 wt.% ZnB7microcrystalline arc-melted Ti18Zr24Nb (at.%)B8hot-pressed at 600 °C for 10 minB9Ti23Zr25Nb-based foam.

### 2.1. Sample Preparation

The production of all materials was conducted from the following powdered precursors: Ti (CAS:7440-32-6, 99.9% purity, Alfa Aesar, Haverhill, MA, USA), Nb (CAS:7446-03-1, 99.8% purity, Sigma Aldrich, St. Louis, MO, USA), 45S5 Bioglass (45% SiO_2_, 24.5% Na_2_O, 24.5 CaO, 6% P_2_O_5,_ Mo-Sci GL0160P, Mo-Sci, Rolla, MO, USA), Ag (CAS:7440-22-4, 99.9% purity, Alfa Aesar) powders, Cu (CAS:7440-66-6, 99% purity, Sigma Aldrich) turnings, Zn (CAS:7440-66-6, 99% purity, Sigma Aldrich) granules, and Zr fillings from a sponge (CAS:7440-50-8, purity ≥99%, Sigma Aldrich). The commercial pure (CP-Ti) microcrystalline titanium (Ti) with purity >99.6% (Grade 2) cut from the rod delivered at the annealed state from Goodfellow (Huntingdon, UK) was used as a control sample for the in vitro cytocompatibility test and the nanoindentation measurements. The mechanical alloying process and the powder consolidation (cold compaction and sintering) were provided as in the previous article of our scientific group [41]. The pellets have *d* = 6 mm in diameter and *h* = 3 mm in height.

The Ti23Zr25Nb alloy was electrochemically modified. Before the modification, the sample was polished with the grinding paper of 600 grit and cleaned with an ultrasonic bath and ethanol. Additionally, to conduct the electrochemical process only on the one sample’s surface, other surfaces were protected by covering with the heat-resistant silicone gasket. The process was divided into two stages. The first stage of electrochemical etching was conducted in the bath containing 1 M H_3_PO_4_ + 2% HF with the use of Solartron 1285 potentiostat (Solartron Analytical, Farnborough, UK) and platinum electrode in the voltage of 10 V vs. OCP for 60 min. The second stage was conducted in the bath containing 0.042 M Ca(NO_3_)_2_ + 0.025 M (NH_4_)_2_HPO_4_ + 0.1 M HCl using the same equipment. Process parameters were −10 V vs. OCP voltage and 60 min.

In contrast to nanoindentation load–depth curves of the Ti23Zr25Nb-9BG composite, the results of Ti23Zr25Nb-based foam and microcrystalline arc-melted Ti18Zr24Nb alloy were additionally presented. The foam was made with the addition of 40% ammonium bicarbonate (Alfa Aesar, 98% purity) (to Ti23Zr25Nb powder ratio). Sintering was conducted in two stages. The first stage was carried out at 175 °C for 2 h in the vacuum. In the second stage, the temperature was set at 1150 °C for 10 h in the protective high-purity argon atmosphere.

### 2.2. Materials Characterization

The electrochemically modified sample’s crystal structure was characterized by the Panalytical Empyrean XRD equipment (Almelo, Netherlands) with the copper (CuKα) radiation source. The following measurement parameters were used: voltage 45 kV, anode current 40 mA, 2 theta range 30–80°, time per step 60.325 s/step, and step size 0.0334°.

The microstructure of the modified materials was revealed with the scanning electron microscope technique. A Mira-3 Tescan microscope (Tescan, Brno, Czech Republic) was used for this purpose. The corrosion properties of the materials were determined with the potentiodynamic tests. A Ringer’s solution and Ag/AgCl reference electrode were used for that purpose with the following testing parameters: −1 V to 2.5 V vs. OCP range and 1 mV/s scan speed. The samples’ surface morphology was investigated using a profiler T8000 (Hommel-Etamic, Villingen-Schwenningen, Germany). EVOVIS software (Hommel-Etamic, Villingen-Schwenningen, Germany) was used to analyze the measured profiles, which allows us to calculate roughness parameters (ISO 25178-2) [49]: arithmetic mean roughness (μm)—R_a_, the maximum height of the profile (μm)—R_t_, ten-point mean roughness (μm)—R_z_. Contact angles (CA) for diiodomethane and glycerol testing fluids and surface free energy (SFE) were calculated using the Kruss-DSA25 instrument and Kruss-Advanced 1.5 software (Krüss, Hamburg, Germany). An ellipse fitting method was applied to calculate the contact angles of materials [50]. The surface free energy was calculated using the Owens, Wendt, Rabel, and Kaelble (OWRK) method [51,52]. Tests were carried out at ambient conditions (23 °C). Multiple measurements of the single 2 μL drop, dosed with the speed of 0.2 mL/min, were handled for 2 s with the probing frequency of 50 fps. Corrosion and wettability results were based on the three repeated measurements for the data analysis and uncertainties calculations. Nanoindentation measurements were conducted with Picodentor HM500 (Fischer Technology Inc., Windsor, CT, USA). The parameters were as follows: 300 mN load applied for 5 s. A Vickers’ tip was used as the indenter for these measurements. The elastic modulus of materials based on the load–depth curves was calculated with the Oliver–Pharr method [53]. Details of the run tests are the same as in the previous research studies [41,42].

### 2.3. In Vitro Biocompatibility Studies

Cultures in a conditioned medium were carried out to investigate the potential cytotoxic effects of the insert’s components that entered the solution (breeding medium). To sterilize the inserts, they were immersed in a 70% solution of the ethanol, diluted with distilled water, and dried in a laminar flow hood with the UV sterilization of each side of the sample for 12 h.

#### 2.3.1. Cell Lines Preparation

Primary cell lines, Normal Human Osteoblasts (NHost, CC-2538), and Human Periodontal Ligament Fibroblasts (HPdLF, CC-7049) were purchased together with a dedicated set of breeding media, respectively: CC-3207 OGM Osteoblast Growth BulletKit (CC-3208 + CC-4193) and CC-3205 SCGM Stromal Bullet CellKit (CC-3204 + CC-4181) at LONZA Group Ltd. (Morristown, NJ, USA). After thawing, the cells were sown into 25 cm^2^ cultured bottles. Cell multiplication was carried out until 80% of the cultured vessels’ surface coverage was obtained by cells, and the cells were transferred to new vessels. Cells were cultured under standard conditions in plastic plates in the Haereus BB16 incubator (Haereus, Hanau, Germany) at 37 °C temperature in an atmosphere of 5% CO_2,_ and increased humidity level of 95%.

#### 2.3.2. Conditioning of Breeding Media

Half of each type of prepared and sterilized inserts was placed (the test surface down) in an unsupplemented medium (1.25 mL of substrate per 1 cm^2^ insert) OBM dedicated to breeding the original line of human osteoblasts (Osteoblasts Basal Medium, #CC-3208; LONZA Group Ltd., Morristown, NJ, USA), the second half in an unsupplemented, dedicated medium for the breeding of human fibroblasts, SCBM (Stromal Cell Basal Medium, #CC-3204; LONZA Group Ltd., Morristown, NJ, USA) Inserts were incubated for 24 h in closed culture bottles, on zirconium balls (ϕ2 mm) at 37 °C, on an orbital shaker (100 rpm). After the incubation, the protected medium was centrifuged 1300× *g* for 10 min and then filtered through a 0.22 mm microfilter (Millipore, Burlington, MA, USA). Then, the substrates were protected in a refrigerator (4 °C) until the growth of cells conditioned in the medium was performed. Directly before the experiment, conditioned substrates were supplemented with dedicated sets of supplements: #CC-4193—fetal bovine serum (FBS), vitamin C. 50 mg/mL, and antibiotic mixture (LONZA Group Ltd.; Morristown, NJ, USA) for OBM medium. Medium SCBM was added with a set of #CC-4181—FBS, rhFGF (Recombinant Human Fibroblast Growth Factor), insulin, and antibiotic mix.

#### 2.3.3. The Cytotoxicity Evaluation of Inserts and Cell Survival by MTS Assay

The inserts tested’s possible cytotoxic effect was evaluated using the CellTiter 96^®^ AQueous Non-Radioactive Cell Proliferation Assay (MTS) (Promega, WI, USA). The lifespan assessment was used to measure the ability of live cells to convert MTS into a colored product—a formazan whose concentration is proportional to the number and metabolic activity of cells. Up to 96-well culture plates were sown with 2.5 × 103 cells of each type. The cells were grown for 24, 72, and 120 h in each cell type and test materials, conditioned, fermented, and medium diluted with the fresh added medium in a ratio of 1:1, at a volume of 300 μL per well. Cell culture in a conditioned medium was carried out in parallel with the culture of cells of a given type, growing in a completely fresh, insulating medium.

At the end of each time interval, an MTS test was performed for both cell types following the manufacturer’s instructions. In short, before the test, the culture medium was listed on the test plate by adding 100 mL of a new portion of the conditioned medium mixture in a ratio of 1:1 with a fresh culture medium and 20 mL of MTS working solution. The cells were incubated with culture conditions 37 °C/95% H_2_O/5% CO_2_ for 1.5 h until the product’s color solution was obtained.

The product’s concentration was measured spectrophotometrically at λ = 490 nm in the ELISA plate reader, MRX Dynex (Chantilly, VA, USA). The measurement was made against a negative control, which was a reaction in wells without cells sown. All farms were conducted in triplicate. For each cell type and conditioned medium, the MTS test results were averaged.

The relative viability of the cells (RVC) was calculated based on the value of absorbance from the equation:RVC (%) = [(a − b)/(c − b)] × 100 (%),(1)
where a—mean absorbance of the tested sample (corrected to specific background), b—mean absorbance of blank control (reaction without the cells), and c—mean absorbance of medium conditioned with a reference sample (microcrystalline Ti, with the correction to specific background).

#### 2.3.4. Cell Culture for Photographic Documentation

In parallel, on 24-hole breeding plates, on glass inserts (ϕ13 mm), osteoblasts and fibroblasts were established in conditioned culture media and prepared for the MTS test. Farms were farmed at 37 °C/95% H_2_O/5% CO_2_ for 24, 72, and 120 h. After this time, the cells were fixed in a 2% solution of glutaraldehyde in PBS. The cells fixed on the inserts were colored with methyl blue solution and rinsed in ddH_2_O. After drying, microscopic preparations were made based on which photographic documentation was made in the 150× magnification using a Nikon microscopic kit and a digital camera (Nikon, Tokyo, Japan).

## 3. Results and Discussion

### 3.1. Structure Properties

The electrochemically etched sample, followed with the electrochemical deposition, leads to the calcium and phosphorus-riched coating formation. This coating’s crystal structure was mainly the calcium hydroxide—Ca(OH)_2_ (Figure 1). The other phases that were identified with the X-ray diffraction include hydroxyapatite—Ca_5_(PO4)_3_(OH) (with peaks of much lower intensity) and Ti(β) (the crystal structure of the base Ti23Zr25Nb material).

Moreover, the previously studied phase composition of biocomposites differs from the single-phase Ti23Zr25Nb alloy. Ti(α), Ti_2_ZrO, and Nb_5_Si_3_P are other phases determining these materials’ improved biological properties. The Ti(β) content is decreased to 65% with 9% 45S5 Bioglass content. Additionally, its lattice constant is expanded to the 3.36–3.37 Å phase with 45S5 Bioglass and the content of Ag, Cu, and Zn [41].

Surface SEM images of Ti23Zr25Nb alloys after electrochemical etching at different magnifications are presented in Figure 2. The electrochemical etching allows the development of the Ti23Zr25Nb alloy surface, further improving the Ca and P ions’ deposition to the material’s surface during the next processing stage. Both pits and pores are visible in the presented micrograph.

The morphology of coating obtained after deposition in 0.042M Ca(NO_3_)_2_ + 0.025M (NH_4_)_2_HPO_4_ + 0.1 M HCl at −10 V vs. OCP voltage and 60 min, with EDS mapping, confirms the higher content of both calcium and phosphorous on the surface, forming the mixture of Ca(OH)_2_ and Ca_5_(PO_4_)_3_(OH) complex presented in Figure 3. Additionally, the tilted-views with the estimated coating thickness close to 50 μm are provided in Figure 4.

### 3.2. Mechanical Properties

The mechanical properties of Ti23Zr25Nb alloys (as well as other TiZrNb alloys) and Ti23Zr25Nb-based composites were already studied in our previous work [41,42,54]. In Figure 5, the Young modulus was referenced to arc-melted Ti18Zr24Nb alloy and Ti23Zr25Nb foam. Ti23Zr25Nb alloys produced with hot pressing has a similar modulus [54] as a similar alloy made with conventional arc melting technology as the reference sample. The conventional method of samples consolidation in two stages, cold pressing and sintering, leads to a further limitation caused by the higher porosity of the produced sample [54]. Ti23Zr25Nb-9BG has been proven to have interesting mechanical properties with an elastic modulus equals to approximately 43 GPa [41], which is also lower than the elastic modulus of the Ti23Zr25 sample with 70% porosities produced with the use of ammonium bicarbonate (approximately 56 GPa). The limitation of the Young’s modulus of TiZrNb and 45S5 Bioglass was done despite lower porosities of the so-produced materials. Both of these materials, Ti23Zr25Nb foam and the Ti23Zr25Nb-45S5 composite, seem to be in the same way interesting in terms of their mechanical properties. Their use in biomedical applications in terms of stress shielding effect limitation [55] is much more suitable than using the non-modified Ti23Zr25Nb alloy with the elastic modulus of approximately 70 GPa and 100 GPa for the cold-pressed and hot-pressed samples, respectively [54]. In both cases, the osseointegration process would be stimulated with the pores’ presence in the produced materials. However, other technologies allow limiting these materials’ modulus even further, such as the gel casting technique, which should be the object of further studies [56].

### 3.3. Surface Properties

The surface roughness of the implant significantly influences cell attachments. Additionally, micro- and nanotopography has played a crucial role in cell proliferation [44,57]. The bulk Ti23Zr25Nb alloy’s roughness before and after electrochemical modification are presented in Figure 6.

Ra, Rt, and Rz are equal to 0.56, 12.72, and 8.06 μm for the bulk non-modified sample, respectively (Table 1). Etching leads to the formation of the pores and numerous pits on the sample’s surface, leading to the enhanced proliferation and growth of cells. During the etching process, large pores are formed on the surface of the sample.

### 3.4. Corrosion and Surface Wetting Properties

Several reports have shown that surface treatment methods can decrease Ti’s corrosion rate in simulated body fluids [58,59]. The methods, including titanium plasma-spraying, grit-blasting, acid-etching, and anodization, or calcium phosphate deposition, can lead to coating with increased osteoconduction. The morphology and the properties of the so modified materials are reviewed by Guehennec et al. [58].

The corrosion resistance after electrochemical modification of the Ti23Zr25Nb alloy is improved (Figure 7). Electrochemical etching leads only to a slight decrease of the corrosion current, which is caused by the formation of some anodic oxides among the etching process previously observed after the same treatment of Ti foil [60] and mechanical alloyed Ti6Al4V alloy [61]. Further improvement is possible after the electrochemical deposition of calcium and phosphate-containing coatings consistent with other studies [62]. All of the potentiodynamic curves (Figure 7) reveal a clear passivation in the anodic part in the whole measurement range. However, better corrosion properties are possible with biocomposites containing 45S5 Bioglass formation, especially doped with silver. These composites were proven to improve the corrosion properties significantly (lower corrosion currents and corrosion potential). The corrosion currents of Ti23Zr25Nb-9BG and Ti23Zr25Nb-9BG-Ag are equal to about 10 × 10^−8^A/cm^2^ in contrast to 10 × 10^−7^A/cm^2^ for the Ca(OH)_2_ + Ca_5_(PO4)_3_(OH) coating [41].

The surface free energy and contact angles for an electrochemically etched and deposited sample were significantly improved compared to the non-modified sample (Table 2). Surface free energy was increased to about 57–58 mN/m after both stages of surface modification. Contact angles were decreased to about 28–33° (for both modification stages) in the case of glycerol and about 48° (for electrochemical etching) and 12° (for electrochemical deposition) in the case of diiodomethane. It makes these materials more hydrophilic than before the modification, which is essential for bone tissue growth [63]. The composites’ contact angles are equal to about 45–62° and 58–78° for diiodomethane and glycerol, respectively. The surface energy is equal to about 35–42 mN/m. These values are similar to that of non-modified Ti23Zr25Nb without vital improvement [41].

### 3.5. MTS Assay

NHost and HPdLF cultivated in the conditioned medium revealed various growth patterns, depending on the sample chemical composition and its microstructure modifications. Additionally, the time of culture influence the increase of the living cells. On the other hand, the growth rate was differentiated between the different cultures (Figure 8 and Figure 9). It is important to note that NHost comparing to HPdLF cells overgrow faster and more regularly on the tested materials.

The MTS assay is used to evaluate the cell proliferation, cell viability, and cytotoxicity of biomaterials. The reduction of the MTS tetrazolium compound by viable cells to generate a soluble in cell culture media colored formazan dye is the MTS assay base. MTS is more efficient than MTT and produces water-soluble formazan that does not require other additives such as DMSO. The quantity of dye product is measured by absorbance at 490 nm wavelength and is directly related to the number of metabolically active cells in culture. A commercial titanium rod was used as the growth control of the cells on the surface of the composite Ti23Zr25Nb—9 wt.% 45S5 Bioglass—type samples. In each culture, cells were grown three times directly on samples of the test materials for 24, 72, and 120 h. To control the natural spread of cells, they were grown directly on culture dishes in the plates without the tested material samples.

The rate of the viability of cells cultured directly on the surfaces of the studied materials is the measurement of the cytotoxicity of TZN-based materials. MTS results showed (Figure 10 and Figure 11) a varying influence of the tested materials on the NHost and HPdLF cells’ viability. Both types of cells showed reduced viability in comparison to the blank cultures. No cytotoxic effect of the samples had been perceived both on the commercial Ti rod and tested Ti23Zr25Nb alloy and composites. However, the intensity of the cell growth of fibroblasts and osteoblasts is heavily reliant on the composites’ chemical composition and surface morphology. The experimental results show that the untreated as well as the electrochemically etched and Ca(OH)_2_ + Ca_5_(PO4)_3_(OH) deposited samples on the 5th day of cell proliferation exceeded the rate of the cell proliferation in the medium conditioned with microcrystalline Ti.

The proliferation of the fibroblasts and osteoblasts cells in conditioned mediums is expressed as a percent of the relative viability potential (RVC) value of the reference medium, conditioned with a CP-pure Ti sample. After 72 h, the proliferation of both cells in the media conditioned with all tested samples exceeded the cell proliferation rate compared to the reference pure Ti specimen. Despite the natural diversities between the NHost and HPdLF cell lines, the tested cells showed similar RVC in contact with tested samples compared to the control. While the vitality rate of the cells cultured directly on the surface of the modified materials exceeded the vitality of the cells growing on Grade 2 Ti, particularly good results were observed for the B2 (Ti23Zr25Nb—9 wt.% 45S5 Bioglass), B3 (Ti23Zr25Nb—9 wt.% 45S5 Bioglass—1 wt.% Ag), B4 (Ti23Zr25Nb—9 wt.% 45S5 Bioglass—1 wt.% Cu) and B5 (Ti23Zr25Nb—9 wt.% 45S5 Bioglass—1 wt.% Zn) samples.

In our earlier study, the biocompatibility of the samples of anodized Ti modified with deposited Ag nanodendrites was investigated [64]. The assessment of fibroblasts and osteoblasts growth served as a model of the soft and hard tissue cells’ performance in contact with the titanium-based materials. The results clearly show that the modification of titanium with silver improves the biocompatibility rate compared to untreated samples. A higher cell RVC was obtained for the cells cultured modified materials than for cells in contact with the control samples’ surface. Modification of the titanium conducted with the anodic oxidation process at high voltages followed by the deposition of silver nanodendrites significantly changed the material’s properties, leading to improved biocompatibility [64].

Silver possesses numerous benefits from the viewpoint of biomedical applications, such as good antibacterial properties, exceptional biocompatibility, and acceptable stability [65]. The addition of Ag into bioactive glasses composition has been studied by incorporating Ag ions into BG particles, which does not influence the biocompatibility of BG, but it improves the antibacterial properties of materials [66,67]. The addition of silver can also considerably decrease the bacterial *S. mutans* and *S. aureus* on the bulk Ti-10 wt.% 45S5 Bioglass-1.5 wt.% Ag surface compared to that on the pure Ti rod surface [68]. The antibacterial activity of TiMo–HA composite modified with silver, tantalum (V) oxide, or cerium (IV) oxide against *Staphylococcus aureus* was investigated. The composites modified with Ag and CeO_2_ possess significantly lower adhesion levels of *S. aureus* (p < 0.05). In general, the Ag ions provide both bacteriostatic and bactericidal effects for the *E. coli* and Gram (+) and Gram (−) bacteria without harmful effects to the human cells [69].

The antibacterial effect of mechanically alloyed Ti23Zr25Nb-based composites was also evaluated. Produced materials doped with Ag, Cu, and Zn have promising high antibacterial activity against *S. mutans* [41]. Those properties among high biocompatibility make these materials attractive in use in the medical field of applications as dental implants.

## 4. Conclusions

In this work, novel ultrafine-grained Ti23Zr25Nb-based materials with 45S5 Bioglass and Ag, Cu, or Zn additions were produced through the combination of mechanical alloying and powder metallurgy. Additionally, Ti23Zr25Nb alloy was electrochemically modified in the two stages of processing: electrochemical etching in the solution of H_3_PO_4_ and HF followed by electrochemical deposition in Ca(NO_3_)_2_, (NH_4_)_2_HPO_4_, and HCl. That studied possesses a Young modulus considerably lower than that of Grade 2 Ti. Furthermore, surface modification significantly improves the corrosion resistance and wettability properties. The biocompatibility MTS assays indicates that the bulk composites based on Ti23Zr25Nb and 45S5 Bioglass, with the addition of Ag, Cu, or Zn are suitable candidates for future dental and orthopedics implants.

## Figures and Tables

**Figure 1 materials-13-05252-f001:**
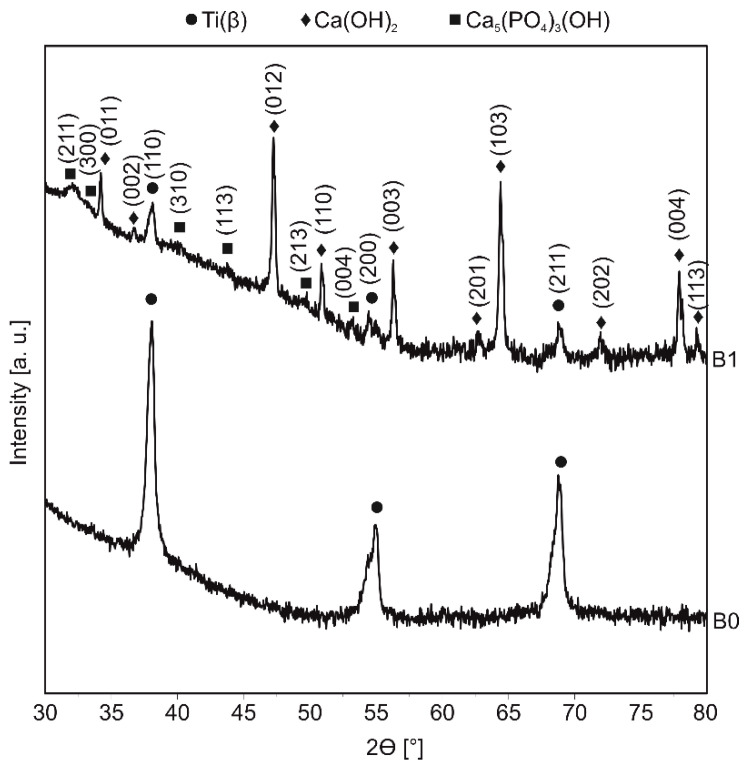
XRD spectra of Ti23Zr25Nb alloy electrochemically modified in 0.042M Ca(NO_3_)_2_ + 0.025M (NH_4_)_2_HPO_4_ + 0.1M HCl bath with the voltage of −10 V and the time of 60 min (B1). The base Ti23Zr25Nb (B0) alloy was used as a reference material.

**Figure 2 materials-13-05252-f002:**
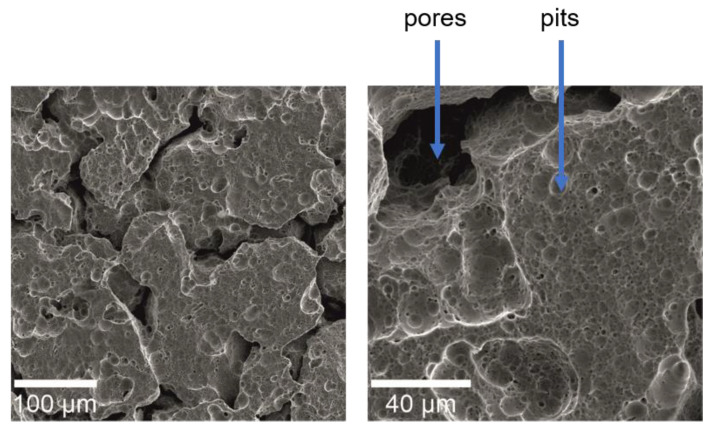
SEM micrographs of Ti23Zr25Nb alloy electrochemically etched in 1 M H_3_PO_4_ + 2% HF in the voltage of 10 V vs. OCP for 60 min (B1*) with different magnifications.

**Figure 3 materials-13-05252-f003:**
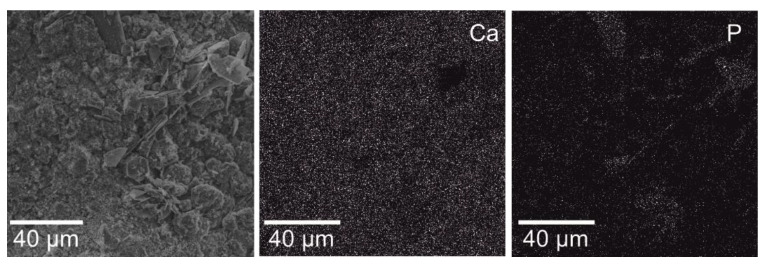
SEM micrographs, EDS mapping of the Ca and P distribution in the electrochemically modified Ti23Zr25Nb alloy in 0.042 M Ca(NO_3_)_2_ + 0.025 M (NH_4_)_2_HPO_4_ + 0.1M HCl bath with the voltage of −10 V and the time of 60 min (B1).

**Figure 4 materials-13-05252-f004:**
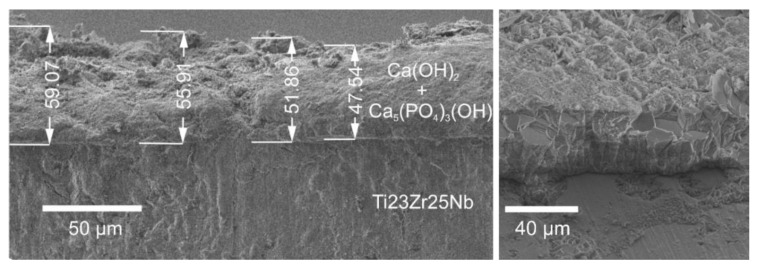
Tilted-views of Ti23Zr25Nb modified in 0.042M Ca(NO_3_)_2_ + 0.025M (NH_4_)_2_HPO_4_ + 0.1 M HCl bath with the voltage of −10 V vs. OCP and the time of 60 min (B1) at 90 (on the left) and 51 (on the right) tilt angle.

**Figure 5 materials-13-05252-f005:**
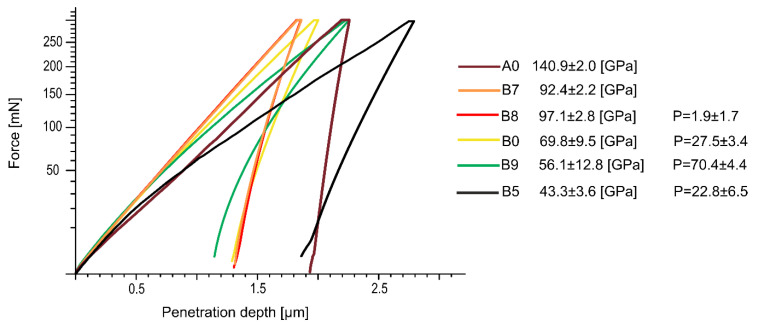
Nanoindentation load–depth curves of Ti23Zr25 alloys produced with hot pressing (B8), cold pressing with sintering (B0), as foams with ammonium bicarbonate (B9), and modified with 9% of 45S5 Bioglass addition (B5). Curves were contrasted with commercially pure Ti (A0) and microcrystalline arc-melted Ti18Zr24Nb (B7). The porosity results based on planimetric analysis (P) of each material were also provided.

**Figure 6 materials-13-05252-f006:**
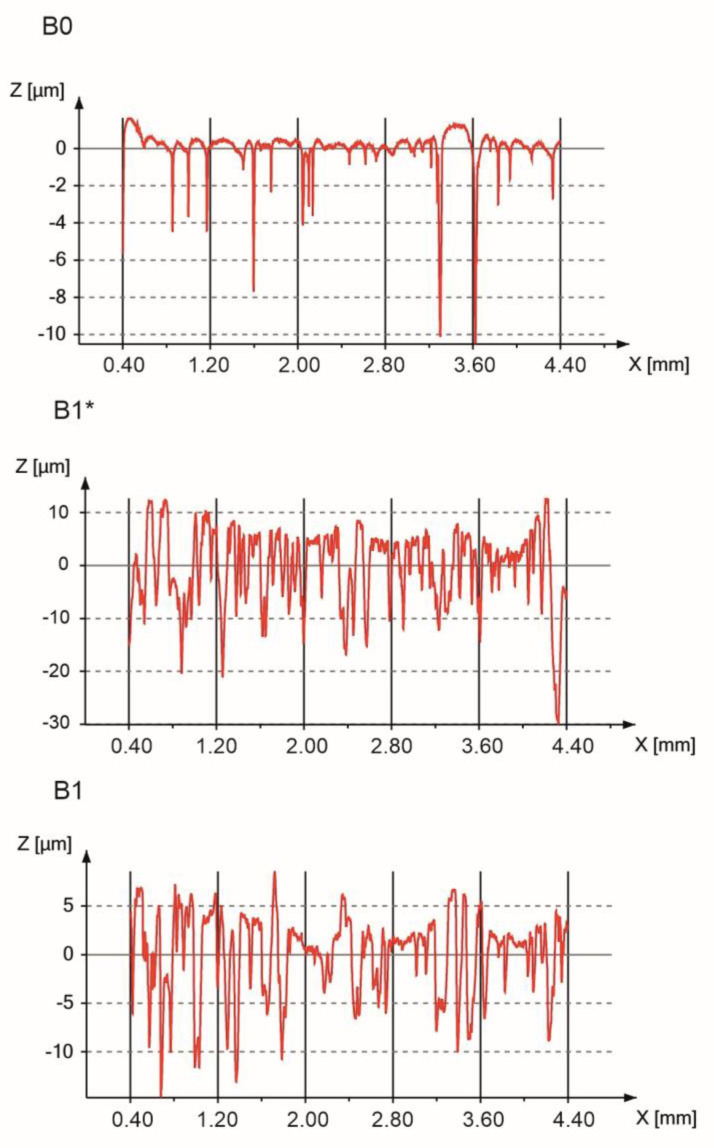
Roughness profiles of the bulk Ti23Zr25Nb alloy before etching (B0), after etching (B1*), and after etching with Ca(OH)_2_ + Ca_5_(PO4)_3_(OH) deposition (B1).

**Figure 7 materials-13-05252-f007:**
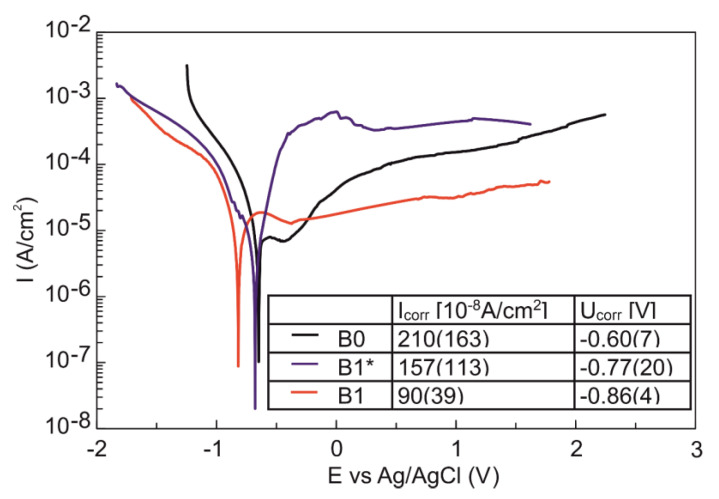
Potentiodynamic curves of Ti23Zr25Nb alloy after electrochemical etching (B1*) and electrochemical deposition (B1) in contrast to non-modified Ti23Zr25Nb alloy.

**Figure 8 materials-13-05252-f008:**
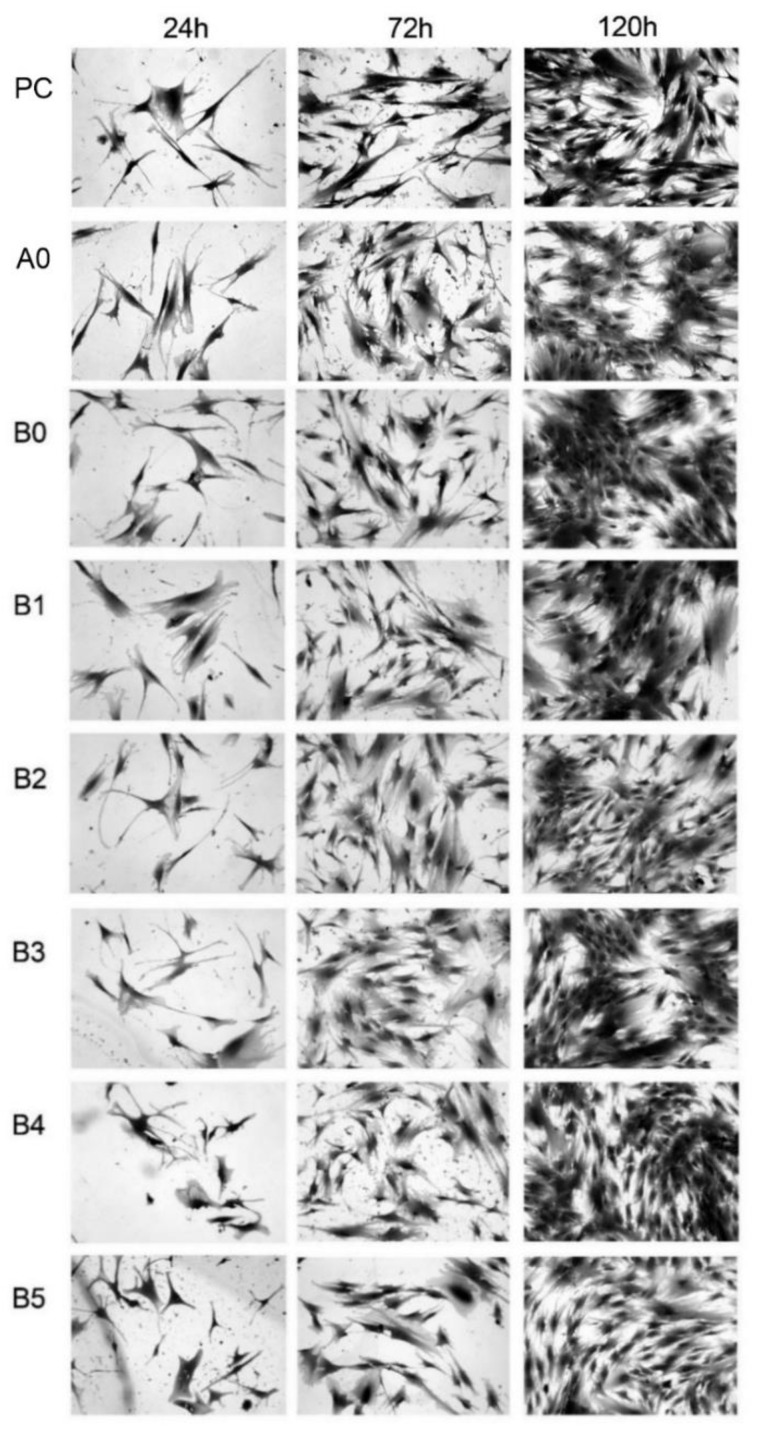
Morphology of the NHost cells cultured in different conditioned media for different times: 24, 72, and 120 h: positive control (PC), microcrystalline Ti (A0), ultrafine grained Ti23Zr25Nb alloy (B0), ultrafine grained Ti23Zr25Nb alloy electrochemically modified (B1) and Ti23Zr25Nb—9 wt.% 45S5 Bioglass (B2), Ti23Zr25Nb—9 wt.% 45S5 Bioglass—1 wt.% Ag (B3), Ti23Zr25Nb—9 wt.% 45S5 Bioglass—1 wt.% Cu (B4), Ti23Zr25Nb—9 wt.% 45S5 Bioglass—1 wt.% Zn (B5) composites.

**Figure 9 materials-13-05252-f009:**
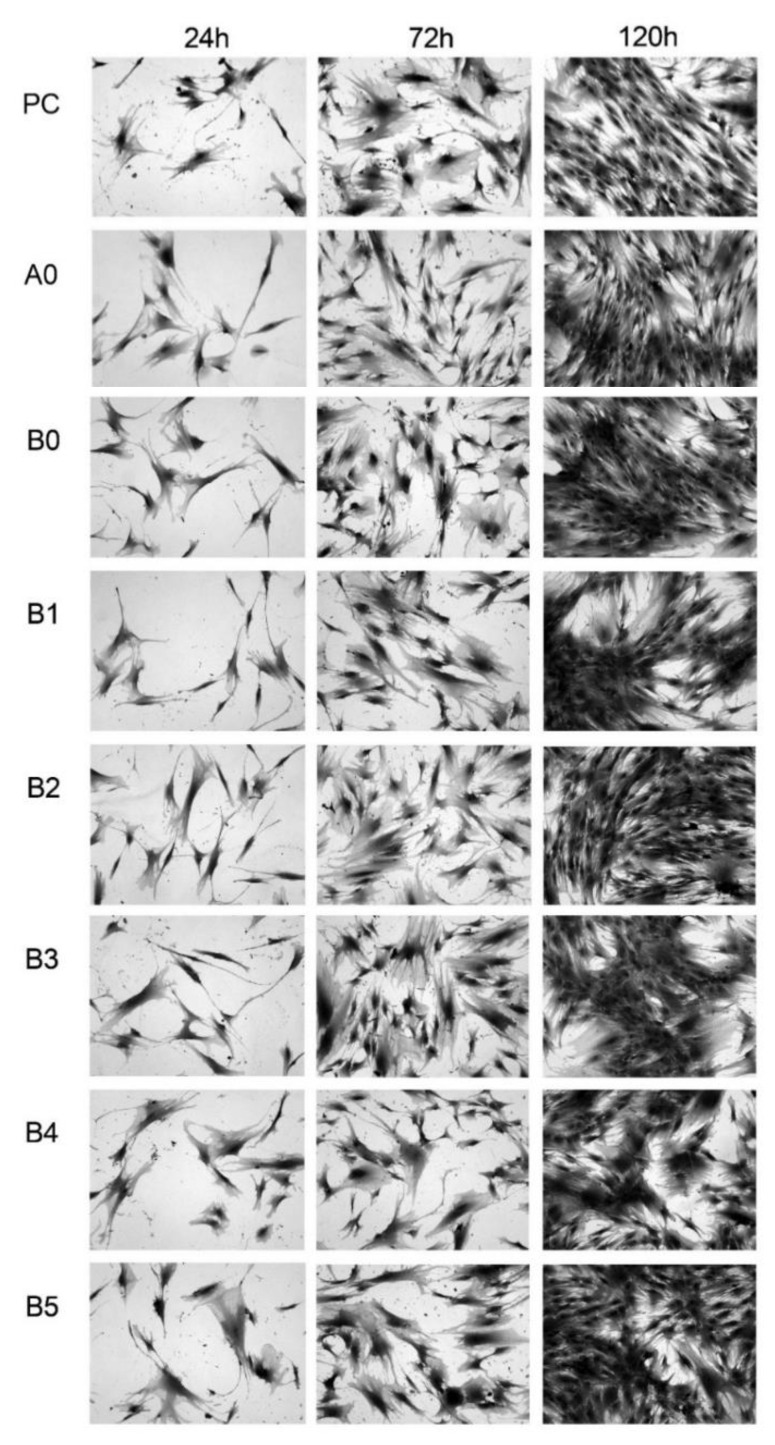
Morphology of the Human Periodontal Ligament Fibroblasts (HPdLF) cells cultured in different conditioned media for different times: 24, 72, and 120 h: positive control (PC), microcrystalline Ti (A0), ultrafine-grained Ti23Zr25Nb alloy (B0), ultrafine-grained Ti23Zr25Nb alloy electrochemically modified (B1), and Ti23Zr25Nb—9 wt.% 45S5 Bioglass (B2), Ti23Zr25Nb—9 wt.% 45S5 Bioglass—1 wt.% Ag (B3), Ti23Zr25Nb—9 wt.% 45S5 Bioglass—1 wt.% Cu (B4), Ti23Zr25Nb—9 wt.% 45S5 Bioglass—1 wt.% Zn (B5) composites.

**Figure 10 materials-13-05252-f010:**
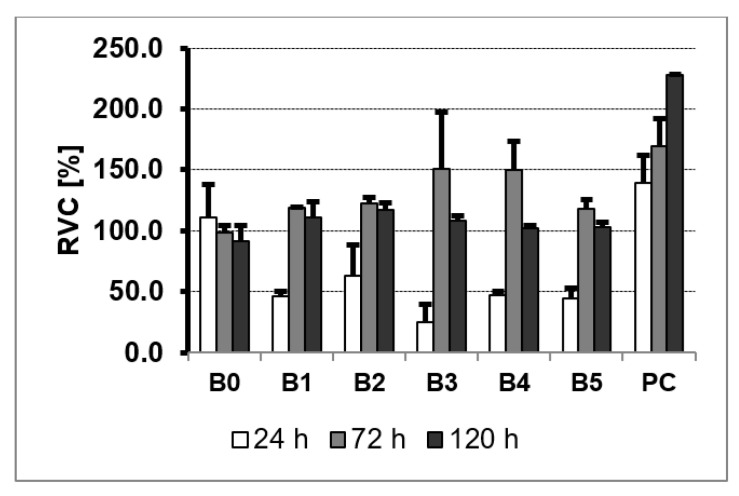
The results of the MTS assays performed at 24, 72, and 120 h on the viability of the NHost: ultrafine-grained Ti23Zr25Nb alloy (B0), ultrafine-grained Ti23Zr25Nb alloy electrochemically modified (B1) and Ti23Zr25Nb—9 wt.% 45S5 Bioglass (B2), Ti23Zr25Nb—9 wt.% 45S5 Bioglass—1 wt.% Ag (B3), Ti23Zr25Nb—9 wt.% 45S5 Bioglass—1 wt.% Cu (B4), Ti23Zr25Nb—9 wt.% 45S5 Bioglass—1 wt.% Zn (B5) composites; PC—positive control.

**Figure 11 materials-13-05252-f011:**
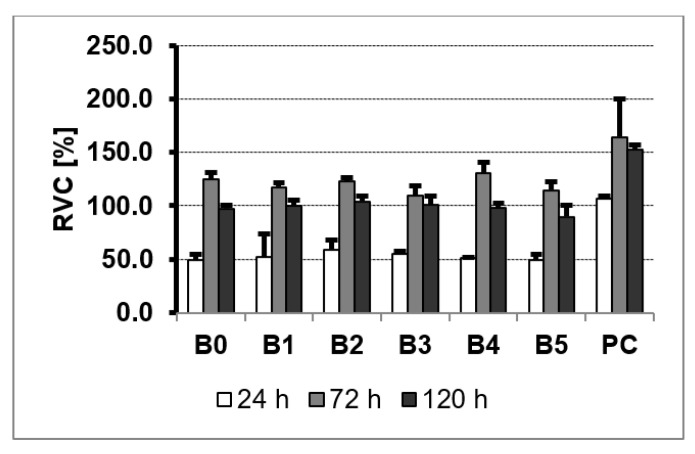
The results of the MTS assays performed at 24, 72, and 120 h on the viability of the HPdLF: ultrafine-grained Ti23Zr25Nb alloy (B0), ultrafine-grained Ti23Zr25Nb alloy electrochemically modified (B1), and Ti23Zr25Nb—9 wt.% 45S5 Bioglass (B2), Ti23Zr25Nb—9 wt.% 45S5 Bioglass—1 wt.% Ag (B3), Ti23Zr25Nb—9 wt.% 45S5 Bioglass—1 wt.% Cu (B4), Ti23Zr25Nb—9 wt.% 45S5 Bioglass—1 wt.% Zn (B5) composites; PC—positive control.

**Table 1 materials-13-05252-t001:** Two-dimensional (2D) (R_a_, R_t_, R_z_) parameters for the Ti23Zr25Nb (at.%) on different processing routes.

Processing Route	R_a_ (µm)	R_t_ (µm)	R_z_ (µm)
B0—polished	0.56 ± 0.11	12.72 ± 1.33	8.06 ± 0.83
B1*	5.58 ± 0.14	39.08 ± 7.02	30.33 ± 4.38
B1	3.42 ± 1.04	27.36 ± 7.67	19.14 ± 4.98

**Table 2 materials-13-05252-t002:** Surface free energy (SFE), dispersive and polar parts of SFE, diiodomethane, and glycerol contact angles for Ti23Zr25Nb alloy after electrochemical etching (B1*) and electrochemical deposition (B1) in contrast to non-modified Ti23Zr25Nb alloy (B0).

	CA (M) Diiodomethane (°)	CA (M) Glycerol (°)	SFE (mN/m)	Disperse (mN/m)	Polar (mN/m)
B0	62.2 ± 9.0	64.6 ± 4.9	35.1 ± 10.0	27.4 ± 5.6	7.7 ± 4.4
B1*	47.6 ± 5.0	27.8 ± 5.9	56.5 ± 8.0	35.6 ± 2.9	20.9 ± 5.1
B1	11.6 ± 8.0	33.4 ± 10.2	58.4 ± 4.3	49.6 ± 1.2	8.8 ± 3.1

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
