# Peer review of "Composite and Surface Functionalization of Ultrafine-Grained Ti23Zr25Nb Alloy for Medical Applications"

_materials, 2020, doi:10.3390/ma13225252_

Round 1

Reviewer 1 Report

Comments to the authors:

  1. Please review the manuscript carefully, make sure no typo and other wrong sentences.
  2.  Figure 5, it's so weird to see the Load-depth curve like this, please make the X-depth, and Y-load. I have never seen the indentation L-D curve like Figure 5, so please change it. 
  3. regarding the indentation test, what tip did you use and why? how did you calculated or observed the Young's modus for these alloys.

Thank you

Reviewer 2 Report

The manuscript "Composite and surface functionalization of ultrafine-grained Ti23Zr25Nb alloy for medical applications" has been reviewed.

The manuscript is clear and correctly set for introduction, materials and method, results and discussion.

However the following aspects should be clarified:

Par. 3.1 - Structure properties: X-Ray source employed for XRD is not indicated. Miller indices of each phase are not reported on Fig. 1.

Par. 3.3 - Adhesion properties after Ca-P deposition should be clarified.

General grammar comment: please check the possessive case in the whole manuscript.

Reviewer 3 Report

Dear author,

I have some comments to your interesting paper.

First, check your English. There are some mistakes, i.e. singular vs. plural (results (pl.)...... confirms (sing.). Also the style could be improved, i.e. ...method for modification properties (preposition missing). I suggest ... method to modify the properties...

Please add the denotations (A0-B9) in the figure captions. It is written in the figures but in some captions it is missing (e.g. Fig. 7).

Did you perform XRD also on the etched surface. It would be intersting to see whether ther is Ti-fluoride in fig 2. Is the image (Fig. 2right) the same surface as image 3left? They look different but the caption says both are B1* but after phosphate it is B1?!

How was the surface finish of your specimens before etching?

How did you manage to apply the phosphate deposition only on one side (Fig. 4). The bare substrate is visible. Did you cut the specimen ür was this surface area covered somehow? 

You should add nanoindentation in fig. 5.

In Fig.6 you should mention that for B1* the y-axis is mm. Both other graphs show µm.

In chapter 3.5 the description is missing. It is all written in 3.6. I suggest to name A0 as a reference and not control. Control is a bare plate. In the figure captions 8 and 9 Ti should be replaced by A0.

What does positive control (PC) mean in Fig. 10, 11? Is this Ctrl in Fig. 8, 9?

Can you commnet on the drop in living cells between 72 and 120 h. Are they dying? The drop is different pronounced for the different materials. Ag seems to be not the best for HPdLF growth. Here Cu seems to be better. Can you comment on that?

Regards

Reviewer
